# High Hole Concentration and Diffusion Suppression of Heavily Mg-Doped p-GaN for Application in Enhanced-Mode GaN HEMT

**DOI:** 10.3390/nano11071766

**Published:** 2021-07-07

**Authors:** Jin-Ji Dai, Thi Thu Mai, Ssu-Kuan Wu, Jing-Rong Peng, Cheng-Wei Liu, Hua-Chiang Wen, Wu-Ching Chou, Han-Chieh Ho, Wei-Fan Wang

**Affiliations:** 1Department of Electrophysics, National Yang Ming Chiao Tung University, Hsinchu 30010, Taiwan; jinjidai@gmail.com (J.-J.D.); maithucs@gmail.com (T.T.M.); wusykuann@gmail.com (S.-K.W.); sakura77988@gmail.com (J.-R.P.); william798424@gmail.com (C.-W.L.); a091316104@gmail.com (H.-C.W.); 2Technology Development Division, Episil-Precision Inc., Hsinchu 30010, Taiwan; hanchiehho@epi.episil.com (H.-C.H.); warren.wang@epi.episil.com (W.-F.W.)

**Keywords:** GaN material, Mg doping, MOCVD, Hall measurement, PL spectroscopy

## Abstract

The effect of Mg doping on the electrical and optical properties of the p-GaN/AlGaN structures on a Si substrate grown by metal organic chemical vapor deposition was investigated. The Hall measurement showed that the activation efficiency of the sample with a 450 sccm Cp_2_Mg flow rate reached a maximum value of 2.22%. No reversion of the hole concentration was observed due to the existence of stress in the designed sample structures. This is attributed to the higher Mg-to-Ga incorporation rate resulting from the restriction of self-compensation under compressive strain. In addition, by using an AlN interlayer (IL) at the interface of p-GaN/AlGaN, the activation rate can be further improved after the doping concentration reaches saturation, and the diffusion of Mg atoms can also be effectively suppressed. A high hole concentration of about 1.3 × 10^18^ cm^−3^ can be achieved in the p-GaN/AlN-IL/AlGaN structure.

## 1. Introduction

The AlGaN/GaN high-electron mobility transistor (HEMT) on Si has received tremendous research attention in high-power device application due to its large breakdown electric field, high electron saturation velocity, and good thermal conductivity [1]. In order to guarantee a safe operation and simplify the circuit architecture, the AlGaN/GaN HEMT is made in the enhanced mode (E-mode) configuration of normally-off operation [2]. The most common and commercial E-mode HEMT is designed in the p-GaN/AlGaN/GaN HEMT configuration. The p-GaN raises the GaN conduction band of the AlGaN/GaN HEMT above the Fermi level, leading to the depletion of the two-dimensional electron gas (2DEG) channel at zero gate bias [3]. Therefore, an E-mode p-HEMT with a higher and stable threshold voltage (Vth) is expected by increasing the hole concentration. However, Mg doping for higher hole concentrations encountered several challenges, including (1) the compensation effect of the donor due to native defects (V_N_) and dislocations [4,5,6], (2) low p-type activation of Mg-H into GaN [7,8], (3) self-compensation effect due to saturation Mg doping-induced donor-type defects [9,10,11], (4) the formation of pyramidal defects from Mg segregation on threading dislocation [12,13], and (5) Mg diffusion into the AlGaN barrier layer and GaN channel layer [14,15]. Although, L. Sang et al. recently showed that the hole concentration and activation efficiency of Mg-doped p-GaN grown on a free-standing GaN substrate of a low dislocation density could be improved dramatically [6]. Yingda Chen et al. discovered that the growth technique of indium surfactant-assisted delta doping could substantially enhance the hole concentration of a p-GaN/u-GaN homo-structure grown on a 2-inch c-plane sapphire to 1.5 × 10^18^ cm^−3^ [16]. However, the issue of low activation efficiency for Mg-doped p-GaN/AlGaN hetero-structures on the more economic Si substrates remains. As the Mg doping increases, the deep-level emission dominates in the photoluminescence (PL) and cathodoluminescence (CL) spectra [17,18]. This implies the formation of deeper donors to compensate holes or the creation of deeper Mg acceptor levels rather than shallow acceptor levels, further resulting in the difficulty to activate holes from the deep Mg acceptors to the valence band and decrease the activation efficiency. Therefore, it is essential to further investigate the effect of Mg doping on the electronic and optical properties to discover the optimized growth condition for better activation efficiency.

With the case of Mg diffusion into the AlGaN barrier layer and GaN channel layer, Loizos Efthymiou et al. discovered that Vth shifts firmly with Mg diffusion [19]. CL measurements revealed Mg diffusion along the dislocation [20]. Mg diffusion along the edge-type and mixed-type dislocations was also evidenced by transmission electron microscopy and atom probe tomography [21,22]. As a result, it is crucial to explore how to suppress Mg diffusion for better device performance of Mg-doped p-GaN/AlGaN/GaN HEMTs.

In the current work, the flow rate of Cp_2_Mg was modulated to grow Mg-doped p-GaN on AlGaN to study the effect of different Mg doping concentrations on the hole concentration and activation efficiency. The PL experiment was carried out to investigate the deep emissions and self-compensation at various doped Mg levels. In addition, Hsien-Chin Chiu et al. demonstrated that a thin AlN etch stop layer in the p-GaN/AlN/AlGaN/GaN HEMT structure can effectively improve the device R_ON_ uniformity and reduce the leakage current [23,24]. Thus, the influence of a thick GaN and thin AlN interlayer (IL) at the interface of the Mg-doped p-GaN and AlGaN layer on the activation efficiency and Mg diffusion was also investigated in this study.

## 2. Materials and Methods

The epitaxial structures of the Mg-doped GaN layers were grown by a metal organic chemical vapor deposition (MOCVD) system (Veeco Instruments Inc, Plainview, NY, USA) on 6-inch Si (1 1 1) substrates, as shown in Figure 1. The conventional source precursors including trimethylaluminum (TMAl), trimethylgallium (TMGa), ammonia (NH_3_), and bis(cyclopentadienyl) magnesium (Cp_2_Mg) were used to grow the AlN, AlGaN, GaN, and Mg-doped p-GaN layers. To avoid Ga-Si melt-back etching, a 200 nm AlN nucleation layer was first grown at 1030 °C on Si substrate. There are three types of samples, A, B, and C, as shown in Figure 1. All samples applied the same step-graded AlGaN buffers, consisting of a 200 nm Al_0.7_Ga_0.3_N layer, a 300 nm Al_0.5_Ga_0.5_N layer, and a 300 nm Al_0.__3_Ga_0.7_N layer grown at 1020 °C to modulate stress for avoiding cracking. The sample structures were designed for high Mg activation rates and suppressing Mg diffusion into the under-layers. For sample A, the 1000 nm-thick Mg-doped p-GaN layers were grown at 990 °C with different Cp_2_Mg flow rates of 0, 200, 450, 600, 750, and 900 sccm, labeled as A_0_, A_200_, A_450_, A_600_, A_750_, and A_900_, respectively. For both samples B and C, the Cp_2_Mg flow rate was 900 sccm for investigating the effect of undoped GaN (u-GaN) and AlN-IL on the Mg activation rate and diffusion. The post-growth thermal activation of Mg-doped p-GaN was performed for 20 min at 720 °C under a nitrogen atmosphere. The secondary ion mass spectroscopy (SIMS) measurement was carried out on all samples to determine the Mg concentration in the p-GaN layer by the IMS-6f (CAMECA SAS, Gennevilliers, France). In order to investigate the electrical properties of p-GaN, the standard Hall effect with the Van der Pauw method was conducted at room temperature by the HMS-3000 (Ecopia Corporation, Anyang-City, South Korea). The optical properties of all samples were studied using low-temperature photoluminescence (PL) spectroscopy by the excitation of a HeCd laser at 325 nm. The threading dislocation density (TDD) was evaluated from the full width at half maximum (FWHM), scanned on GaN (002) and (102) planes by X-ray diffraction (XRD, X’Pert Pro MRD, Malvern Panalytical, Almelo, The Netherlands). The characterization of structure strain was performed by Raman scattering. The effect of the Mg doping concentration on the surface morphology was carried out by scanning electron microscopy (SEM, JSM7001F, JEOL, Tokyo, Japan), optical microscopy (OM, AL100, Olympus Corporation, Tokyo, Japan), and atomic force microscopy (AFM, NT-MDT Spectrum Instruments, Moscow, Russia).

## 3. Results and Discussion

The hole carrier concentration and activation efficiency as a function of Mg doping are revealed in Table 1 and Figure 2. As we can see, the hole concentration increases, corresponding to decreased mobility with the increasing Mg doping. Meanwhile, the resistivity decreases initially and then increases with the Mg doping. The activation efficiency (Mg doping efficiency), which is defined as the ratio of the hole concentration (obtained from Hall measurement) and Mg doping density (measured by SIMS), increases initially and reaches a maximum value of 2.22% at Mg doping of 2.42 × 10^19^/cm^3^ (450 sccm), and then it decreases with the Mg doping. This can be attributed to the Mg saturated concentration of about 2 × 10^19^/cm^3^. Furthermore, the low Mg concentration behavior presented in our samples is similar to that of other reported data [11,25] for a GaN:Mg hetero-epitaxial layer on a sapphire substrate, as shown in Figure 2. However, they all showed constant reversion of the hole concentration after Mg saturation, owing to the self-compensation effect. Even A. Klump et al. applied UV illumination to reduce H passivation and the self-compensation impact on the GaN:Mg films, which was just helpful on the concentration below the Mg saturation. In our case, when the Mg doping is more than the self-compensation onset of 2.42 × 10^19^/cm^3^ (450 sccm), it is worth noting that the activated hole concentrations still rise without a hole concentration reversion. However, the decrease in activation efficiency could be ascribed to the starting existence of high Mg doping-induced defects, for example, the formation of Mg interstitials [9,17], nitrogen vacancy V_N_ [9,26], Mg_Ga_-V_N_ complexes [11,27], and pyramidal inversion domain (PID) defects [28,29]. Another scenario could be the building possibility of Mg-N-Mg clusters. The rising formation probability of Mg-N-Mg double acceptors could split the acceptor level and create deeper acceptor states and further decrease the density of a single Mg shallow acceptor. The deeper acceptor states are not active in creating free holes, leading to lower activation efficiency. For even higher Mg doping concentrations, the possibility to generate Mg_3_N_2_ clusters increases [30,31]. The formation of Mg_3_N_2_ clusters decreases the single Mg concentration, and the energy states of Mg_3_N_2_ clusters are deep levels in the energy gap and do not contribute free holes. With a consistent result, we also obtained precipitation of Mg-rich and pyramid-shaped defects on our SEM and optical microscope images, respectively, after the flow rate of 450 sccm (not shown here). The energy-dispersive X-ray spectroscopy (EDS) analysis also exhibited the Mg content of the 900 sccm sample, about 2.4% on the Mg-rich precipitates and around three times that of the blank background (0.79%). In addition, AFM images show the root mean square (RMS) of the surface roughness increases from 0.49 to 1.75 nm in the 5 μm × 5 μm scan area, while the Mg flow rate increases from 0 to 900 sccm.

The PL spectra of p-GaN films at 10 K with different Mg doping concentrations are shown in Figure 3. The PL of the undoped GaN film shows a sharp near-band edge emission (NBE) at 3.46 eV (358.4 nm), as shown in Figure 3a. The broad emissions below 3.2 eV are attributed to the defect emissions from the AlGaN layers. In addition, the oscillation in the PL intensity below 3.2 eV is due to the Fabry–Perot interference of the whole sample structure. By measuring the energy separation of the two nearest peaks ΔE, the total sample thickness could be evaluated by d = hc/(2nΔE) ≈ 2 μm, where h, c, and n are the Planck constant, speed of light, and refraction index at the emission peak, respectively. When the Mg doping is turned on at 200 sccm, the native donor (V_N_) [4] and shallow Mg acceptor pair (DAP) emission dominates the PL spectrum, as it can be seen in Figure 3b. The peak of DAP is around 3.1 eV. As the Mg doping is further increased to 450 sccm, the peak energy of blue luminescence (BL) is near 2.8 to 3.0 eV (Figure 3c). The emission peak near 2.8–3.0 eV was attributed to the deep donor-to-shallow acceptor transition [32,33]. These deep donors could be created by the heavy Mg doping-induced defects. The emission peak near 2.8–3.2 eV could also be ascribed to the recombination of a native donor and heavy Mg doping-induced deep Mg acceptor. The PL spectra presented in Figure 3d–f, for higher Mg doping samples, are basically the same as in Figure 3c. The peak intensity of green luminescence (GL) and yellow luminescence (YL) becomes more prominent with increased Mg doping, which means that structural defects related to V_N_ begin to increase [26,34]. In general, the collected PL data corroborate the results of electrical measurements mentioned above (Table 1 and Figure 2). As the Mg doping exceeds 450 sccm, the more Mg atoms incorporated into the GaN crystal generate not only more single Mg shallow acceptors but also more Mg-N-Mg deep acceptors, or donor-type defects, leading to a drop-off in activation efficiency. If the BL emission at 2.8–3.0 eV of Figure 3c is due to the donor-to-deep acceptor recombination, the deep acceptors are about 300 to 500 meV above the valence band compared with the activation energy of the shallow acceptor of about 200 meV [35,36]. Therefore, the deep acceptors have lower efficiency to be activated to offer free holes in the valence band for conducting. Similar competition of two emissions was also discovered recently by Hanxiao Liu et al. for their low- and high-Mg doping samples [18]. They attributed the two emissions at 3.25 eV and 2.9 eV to the shallow donor-to-acceptor and deep donor-to-acceptor transitions, respectively. We suggest that the BL near 2.9 eV can be caused by both the deep acceptors and deep donors. The deep acceptors should result from the Mg-rich and Mg_3_N_2_ precipitates to decrease the activation efficiency. The deep donors of donor-like defects from the V_N_ and Mg-V_N_ complexes can decrease the hole concentration by the self-compensation effect.

In order to investigate the effect of Mg diffusion on the activation efficiency, the electrical properties of samples B and C are discussed. Figure 4a shows the SIMS of samples A_900_, B, and C. Mg diffusion is the strongest for sample B, the p-GaN homo-epitaxy of the 200 nm GaN template. The difference in Mg diffusion for samples A_900_ and C is not significant. However, the hole concentration and activation efficiency are very different for three samples, as shown in Figure 4b. Suppose that the hole concentration evaluated by the Hall measurement is majorly contributed by the top part of the p-GaN layers; the similar Mg doping concentrations at the top of the p-GaN layers for all three samples imply that the self-compensation effects are different. We would like to emphasize that the activation efficiency was effectively increased by decreasing the self-compensation effect, while the decrease in Mg diffusion was trivial, as extracted from the SIMS results. The p-GaN film grown on AlN-IL (2 nm)/Al_0.3_Ga_0.7_N has the best activation efficiency of 2.2%. In the event of p-GaN grown on Al_0.3_Ga_0.7_N and GaN, the activation efficiencies are 1.4% and 0.8%, respectively. This could be due to the strain between the layers to suppress the formation of the Mg doping-induced donor-type defects. These results indicate that aluminum has a smaller atomic radius than gallium, which can inhibit Mg diffusion and increase the compressive stress on the GaN:Mg film [37]. It is expected that a high Al composition could significantly suppress the self-compensation effect, reduce the Mg diffusion concentration, and further increase the hole concentration and activation rate.

Many research groups also investigated the role of stable and metastable Mg-H complexes on the activation efficiency [7,8,25]. They discovered that the hole concentration is proportional to the density of H atoms from the Mg-H complex measured by SIMS before thermal annealing. The Mg atoms without the formation of the Mg-H complex could occupy the interstitials, Mg-V_N_ complexes, or lattice positions of nitrogen (Mg_N_). They are donor-type defects and play the role of self-compensation. As shown in Figure 5a, a higher H concentration was observed before annealing in p-GaN/AlGaN and p-GaN/AlN-IL structures than that in the p-GaN/GaN-IL structure. However, the p-GaN/AlN-IL structure displayed a similar H concentration to p-GaN/AlGaN, which could not expound the higher hole concentration and activation efficiency with AlN-IL. Therefore, these two structures were measured by the HRXRD rocking curves for the FWHM of GaN (002) and (102) planes to calculate the threading dislocation densities (TDDs) [38]. The GaN (002)/(102) planes of 678/1024 arcsecs without AlN-IL, respectively, correspond to the screw/edge-type TDDs of 9.24 × 10^8^ and 3.13 × 10^9^ cm^−2^. The screw/edge-type TDDs of 7.98 × 10^8^ and 3.51 × 10^9^ cm^−2^ with the AlN-IL structure were calculated by the GaN (002)/(102) planes of 630/1028 arcsecs. In contrast to the relationship, the total TDDs with AlN-IL slightly increased from 4.05 × 10^9^ to 4.30 × 10^9^ cm^−2^, indicating that the TDDs do not dominate the hole concentration in this case. We recommend excluding the effect of Mg-H and TDDs on the increasing activation efficiency after the Mg doping concentration reaches saturation. Furthermore, in Figure 5b, the PL spectra of p-GaN exhibit that the photon intensities of BL, GL, and YL decreased dramatically with AlN-IL. The lower concentration of self-compensation defects in the p-GaN on AlN-IL could be due to the greater compressive strain in p-GaN. This is consistent with our Raman spectra, where GaN E_2_ (High) and A1 (LO) shift from 563.46 to 563.74 cm^−1^ and 722.26 to 726.19 cm^−1^, respectively. The Raman energy blue shift implies greater compressive stress in the p-GaN epilayer with AlN-IL [39,40]. This effect is in agreement with the suppression of donor-like defects under greater compressive strains from inserting an AlN interlayer into the Mg-doped GaN/AlGaN superlattice by Hu et al. [41,42]. Herein, we would like to express that the existence of the greater compressive stress of heavily Mg-doped GaN is crucial in affecting the self-compensation effect because it can effectively extend the Fermi energy and consequently increase the formation energy of self-compensation defects [9]. This was mentioned in other research [10,43,44] which found that a strain state from compressive to tensile is accompanied by the BL emission due to large local lattice relaxations by the generation of self-compensation defects. This study reveals that a high-Al composition layer under the p-GaN layer can effectively enhance the hole concentration and significantly reduce the self-compensation effect. Furthermore, no reversion of the hole concentration could be observed after Mg saturation. This finding is precious for application in E-mode GaN HEMTs.

## 4. Conclusions

In this study, the flow rate of Cp_2_Mg was modulated to grow heavily Mg-doped p-GaN on AlGaN for application in enhanced-mode HEMTs. A maximum activation rate of 2.22% was accomplished with Mg doping of around 2.42 × 10^19^ cm^−3^. The further increase in the hole concentration with the increasing Mg concentration reveals that the hole reversion could be restrained, owing to the decreased compensation-type defects resulting from the enhanced compressive strain. In addition, a high hole concentration of 1.3 × 10^18^ cm^−3^ with a high activation efficiency was also achieved by heavy Mg doping of around 6.05 × 10^19^ cm^−3^ in the p-GaN/AlN-IL/AlGaN structure. The diffusion of Mg can be effectively suppressed by inserting an AlN layer at the interface of Mg-GaN and AlGaN. The current results provide important information for the growth of Mg-doped p-GaN of a high hole concentration in E-mode HEMT application.

## Figures and Tables

**Figure 1 nanomaterials-11-01766-f001:**
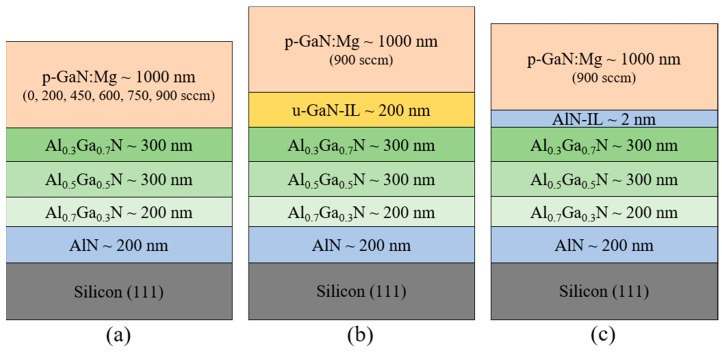
Schematic structures of p-GaN samples: (**a**) sample A with varying Cp_2_Mg flow rates (0, 200, 450, 600, 750, 900 sccm), (**b**) sample B with Cp_2_Mg 900 sccm and an additional 200 nm undoped GaN interlayer, and (**c**) sample C with Cp_2_Mg 900 sccm and a 2nm AlN interlayer.

**Figure 2 nanomaterials-11-01766-f002:**
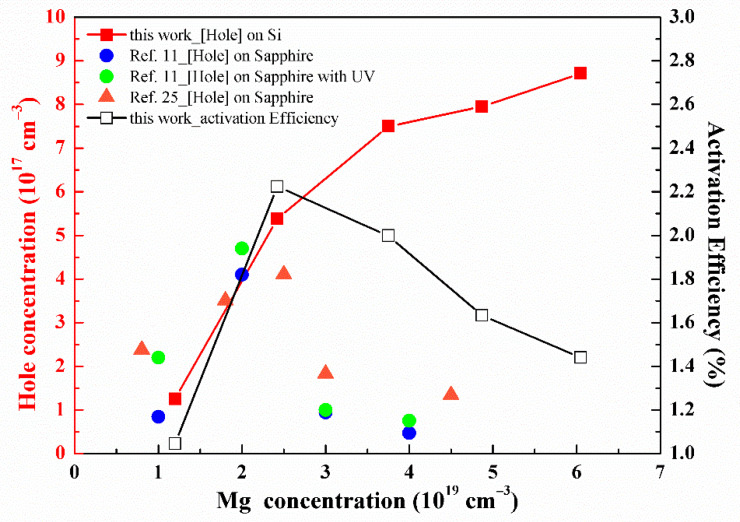
The hole concentration (red solid squares) and activation efficiency (open squares) of p-GaN:Mg layer as a function of the Mg doping concentration. The results from references [11,25] are also plotted for comparison, as shown by blue circles, green circles, and orange triangles.

**Figure 3 nanomaterials-11-01766-f003:**
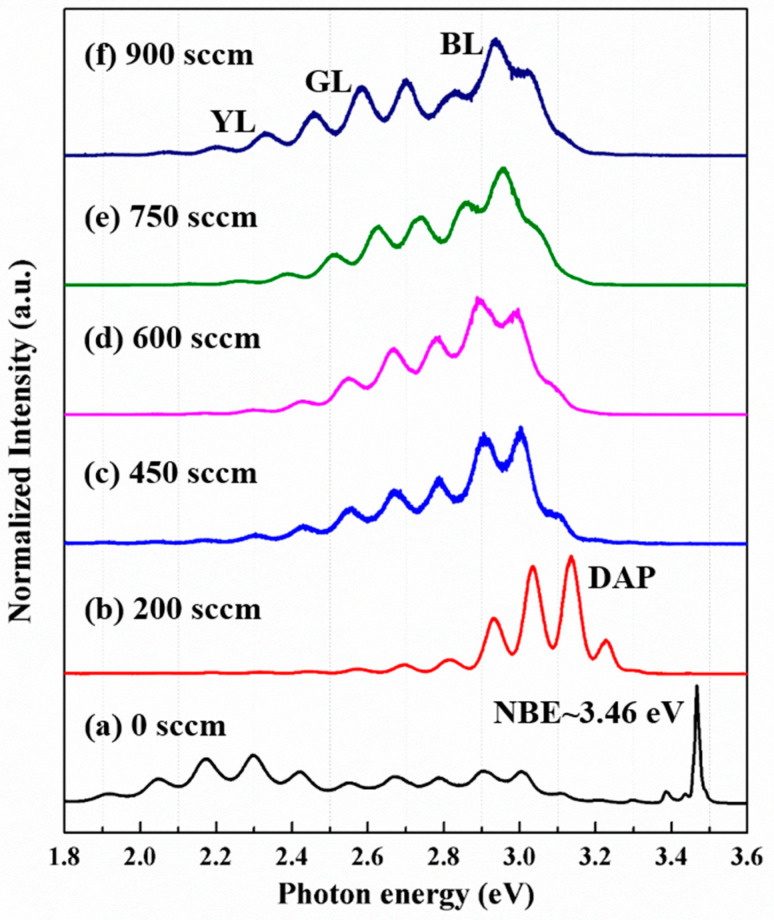
The low-temperature (10 K) PL spectra of p-GaN/AlGaN structures with varying Cp_2_Mg flow rates.

**Figure 4 nanomaterials-11-01766-f004:**
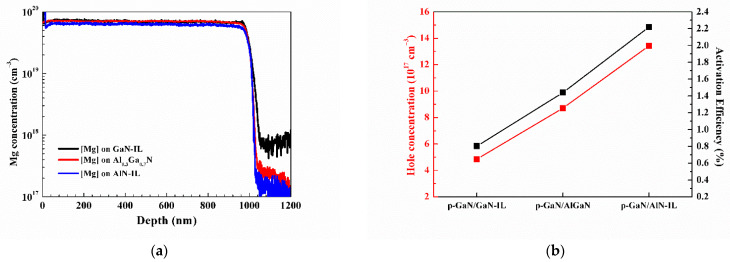
(**a**) Mg concentration profile measured by SIMS for three different structures. (**b**) The hole concentration (red squares) and activation efficiency (black squares) of Mg-doped p-GaN layers grown on GaN-IL, AlGaN, and AlN-IL under-layers.

**Figure 5 nanomaterials-11-01766-f005:**
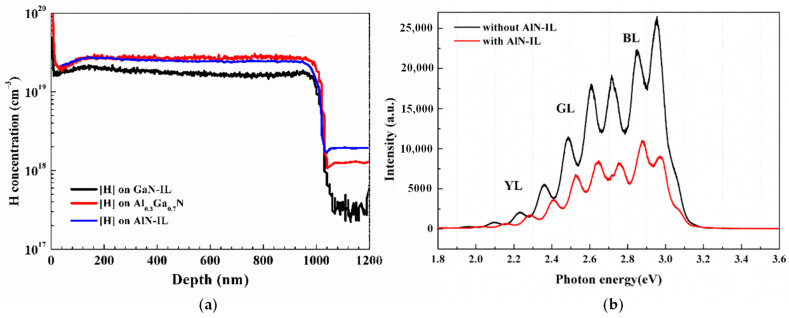
(**a**) The dependence of hydrogen concentration on depth investigated by SIMS. Black, red, and blue lines are results for p-GaN on GaN-IL, Al_0.3_Ga_0.7_N, and AlN-IL structures, respectively. (**b**) The low-temperature (10 K) PL spectra of p-GaN with and without AlN-IL.

**Table 1 nanomaterials-11-01766-t001:** Dependence of activation efficiency and electrical properties on the Cp_2_Mg flow rates.

Structure	Cp_2_Mg Source (sccm)	Mg Doping Concentration (cm^−3^)	Hole Concentration (cm^−3^)	Mobility (cm^2^/V-s)	Resistivity (ohm-cm)	Activation Efficiency (%)
p-GaN/Al_0.3_Ga_0.7_N	200	1.20 × 10^19^	(1.25 ± 0.06) × 10^17^	27.54 ± 1.38	1.87 ± 0.09	1.04
	450	2.42 × 10^19^	(5.38 ± 0.27) × 10^17^	7.69 ± 0.38	1.51 ± 0.08	2.22
	600	3.75 × 10^19^	(7.49 ± 0.37) × 10^17^	5.63 ± 0.28	1.48 ± 0.07	2.00
	750	4.87 × 10^19^	(7.95 ± 0.40) × 10^17^	4.51 ± 0.23	1.74 ± 0.09	1.63
	900	6.05 × 10^19^	(8.71 ± 0.44) × 10^17^	3.54 ± 0.18	2.02 ± 0.10	1.44

## Data Availability

Data are contained within the article.

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
