# Peer review of "High Hole Concentration and Diffusion Suppression of Heavily Mg-Doped p-GaN for Application in Enhanced-Mode GaN HEMT"

_nanomaterials, 2021, doi:10.3390/nano11071766_

Round 1

Reviewer 1 Report

The manuscript describes the effect of Mg doping on the electrical and optical properties of the p-GaN/AlGaN structures on Si substrate grown by MOCVD. The synthesized materials were properly characterized. The interpretation of the data was rational. So I would like to recommend the publication of this manuscript without revision. 

Reviewer 2 Report

This manuscript reports the investigation on Mg-doped p type GaN grown on Si substrate by MOCVD, focusing on hole concentration and activation efficiencies of Mg-doped material. Mg doping is the dominant method used for p-type doping on GaN and there are a large number of reported works that can be found in literatures. Mg-doping is worth studying and the manuscript does provide some valuable results that are interesting. However, the manuscript shows obvious weaknesses in the following aspects.

  • A high hole concentration of about 1.3×1018 cm−3 is claimed in the manuscript with one data point (other data points show 1017 cm−3). I think the authors should explain the stability and repeatability of the result.
  • There are a few literatures reported a hole concentration of around 1.5×1018 cm-3 and a much higher doping efficiency of 12% than the value reported in this manuscript (Yingda Chen et al 2013 Phys. Express 6, 041001(2013)).
  • Frankly, the research methods and technical routes described in the manuscript are not very innovative, although the characterizations of the samples do show some valuable results.

Reviewer 3 Report

   In the proposed manuscript the authors have designed three structures to investigate the Mg activation efficiency and tried to make clear the cause of higher activation efficiency (2.2%) in the p-GaN on AlN-IL. Though analysis of the PL, XRD, SIMS, and Hall measurement data, the author finally attribute to the suppression of donor-like defect under more compressive strains from inserting an AlN interlayer under the p-GaN layer. This conclusion cannot be sufficiently supported by the measurement results. The XRD and SIMS data can indeed exclude the effect of Mg-H and TDDs on the increasing activation efficiency after Mg doping concentration reaches saturation, and the PL data suggest lower concentration of self-compensation defects in the p-GaN on AlN-IL. However, no evidence supports more compressive strain in the p-GaN on AlN-IL. Theoretically, 2 nm AlN-IL cannot induce enough compressive strain in the 1000 nm thick p-GaN. Accordingly, I cannot recommend this manuscript version publish on this journal. Two aspects should make clear:

  1. Evidences to support “more compressive strain in the p-GaN on AlN-IL, e.g., Raman shift.
  2. Exclude the possible effect of two dimensional hole gas (2DHG) in the p-GaN/Al(Ga)N-IL interface, which are widely reported in literatures.

And another suggestion:

The title is “High hole concentration and diffusion suppression of heavily Mg-doped p-GaN for the application in enhanced mode GaN HEMT”, and the authors also declaim that “This study reveals that the introduction of an etch-stop layer with a high Al composition layer under the p-GaN layer not only could improve the device performance of Vth uniformity and current leakage but also increases the hole concentration and suppresses the Mg diffusion.” However, there is no data about the GaN HEMT performance.

Reviewer 4 Report

Please accept all corrections left after first revision.

Round 2

Reviewer 2 Report

The manuscript has been properly modified. But I think the paper by Chen  (Yingda Chen et al 2013 Appl. Phys. Express 6, 041001(2013)) should be cited and, then, the difference, including sampel structrure, should be addressed, which will made the manuscript better.

Reviewer 3 Report

This version can be accepted.

Author Response

This manuscript is a resubmission of an earlier submission. The following is a list of the peer review reports and author responses from that submission.

Round 1

Reviewer 1 Report

The authors investigated the p-GaN doping concentration and activation, which was grown on different starting layers. The authors observed higher activation for p-GaN using an AlN interlayer. The authors suggested that the higher activation was attributed to more Mg-H complex and low compensation type defects and put a reference. While the discussion can make sense, the claim was not validated by scientific analysis. In order to clarify the authors’ conclusion, more experiments involving analysis must be provided. The current manuscript sounds an experiment report due to a lack of scientific analysis.

In conclustion, the authors wrote that this would be useful information for E-mode GaN growth. The authors must consider that when the AlN interlayer is used before p-GaN growth, the polarization effects will cause negative shift in threshold voltage counteracting with the proposed effects. Typical thickness of AlGaN barrier for E-mode structure is about 10 nm or so. When the AlN is ~2 nm, the AlGaN barrier under the AlN layer must be extremely thin, which will cause serious problems in processing and reliability. While the observation might be interested, the potential applications must be presented from the practical point of view.

Reviewer 2 Report

I have read the manuscript and I did not see anything offensive or obviously wrong. The flow is OK and the English is OK as well. The authors discuss several pathways to improve Mg doping in GaN and HEMTs on Si.

They present a comprehensive study and conclusive results. No major concerns were found. 

The significance of the content, quality of presentation and scientific soundness is good. The originality may should be emphasized more in the maintext.

English language and style are fine, just minor spell check required.

Round 2

Reviewer 1 Report

My previous review comments have not been addressed sufficiently in revision. All discussions made in the manuscript are based on other references without providing the authors' own data to support the idea. Unless the authors are able to present new findings or perform scientific analysis to support their findings, this manuscript itself is not enough to be accepted as it is. If the authors had difficulty to perform additional research to support their findings, I suggest that the authors apply their findings for device implementation to validate the idea, e.g. demonstration of high positive Vth or so. This would be definitely suitable for publication.